# A Simultaneous Extraction/Derivatization Strategy for Quantitation of Vitamin D in Dried Blood Spots Using LC–MS/MS: Application to Biomarker Study in Subjects Tested for SARS-CoV-2

**DOI:** 10.3390/ijms24065489

**Published:** 2023-03-13

**Authors:** Yashpal S. Chhonker, Nusrat Ahmed, Christine M. Johnston, Ruanne V. Barnabas, Daryl J. Murry

**Affiliations:** 1Clinical Pharmacology Laboratory, Department of Pharmacy Practice and Science, University of Nebraska Medical Center, Omaha, NE 68198, USA; 2Department of Medicine, University of Washington, Seattle, WA 98101, USA; 3Division of Infectious Diseases, Massachusetts General Hospital, Boston, MA 02114, USA; 4Department of Medicine, Harvard Medical School, Boston, MA 02115, USA; 5Fred and Pamela Buffett Cancer Center, University of Nebraska Medical Center, Omaha, NE 68198, USA

**Keywords:** vitamin D, LC–MS/MS, SARS-CoV-2(+), biomarker

## Abstract

Vitamin D plays a critical role in bone development and maintenance, and in other physiological functions. The quantitation of endogenous levels of individual vitamin D and its metabolites is crucial for assessing several disease state conditions. With cases of severe acute respiratory syndrome coronavirus 2 (SARS-CoV-2) leading to the coronavirus disease 2019 (COVID-19) pandemic, there are several studies that have associated lower levels of serum vitamin D with severity of infection in COVID-19 patients. In this context, we have developed and validated a robust LC–MS/MS method for simultaneous quantitation of vitamin D and its metabolites in human dried blood spot (DBS) obtained from participants tested for COVID-19. The chromatographic separation for vitamin D and metabolites was performed using an ACE Excel C18 PFP column protected with a C18 guard column (Phenomenex, Torrance, CA, USA). The mobile phase consisted of formic acid in water (0.1% *v*/*v*) as mobile phase A and formic acid in methanol (0.1% *v*/*v*) as mobile phase B, operated at a flow rate of 0.5 mL/min. Analysis was performed utilizing the LC–MS/MS technique. The method was sensitive with a limit of quantification of 0.78 ng/mL for all analytes, and had a large dynamic range (200 ng/mL) with a total run time of 11 min. The inter- and intraday accuracy and precision values met the acceptance criteria per the US Food and Drug Administration guidelines. Blood concentrations of 25(OH)D3, vitamin D3, 25(OH)D2, and vitamin D2 over a range of 2–195.6, 0.5–121.5, 0.6–54.9, and 0.5–23.9 ng/mL, respectively, were quantified in 909 DBS samples. In summary, our developed LC−MS/MS method may be used for quantification of vitamin D and its metabolites in DBS, and may be applied to investigations of the emerging role of these compounds in various physiological processes.

## 1. Introduction

Vitamin D plays a critical role in bone development and maintenance, and in other physiological functions [1,2]. In nature, it exists in two major forms: vitamin D2 (ergocalciferol) and vitamin D3 (cholecalciferol). Vitamin D3 can be synthesized endogenously in the human skin and is thus considered the most natural form. Vitamin D2 cannot be made by the human body, partially explaining its low level in the body. The inactive forms of vitamin D (ergocalciferol and cholecalciferol) become biologically active after undergoing two hydroxylation reactions in the liver and kidney, respectively. The first biotransformation reaction occurs in the liver where vitamins D2 and D3 are converted by the hydroxylation reaction into corresponding 25-hydroxy metabolites: 25-hydroxyvitamin D2 [25(OH)D2] or 25-hydroxyvitamin D3 [25(OH)D3]. The reaction is catalyzed by two isoforms of microsomal cytochrome p450 (CYP27A1 and CYP2R1). In the next step, in the renal proximal tubule of the kidney, corresponding 25-hydroxy metabolites of vitamins D2 and D3 are converted into 1,25-dihydroxy or 24,25-dihydroxy metabolites by the action of microsomal enzymes CYP27B1 or CYP24A1, respectively [3,4]. Among these two possible dihydroxy metabolites of vitamin D, 1,25(OH)_2_D is the principal biologically active metabolite of vitamin D. For bone formation and mineralization, the level of 1,25(OH)_2_D is crucial as it regulates the plasma calcium level within the physiological range [5]. Here, it should be noted that although 1,25(OH)_2_D shows the biological actions in the body, quantitating the level of 1,25(OH)_2_D is not the best indicator of vitamin D status in some physiological conditions [2]. Hence, for the clinical diagnosis and monitoring of vitamin D status, 25(OH)D level has most commonly been reported.

The relation between low vitamin D level with various disease conditions including cancer risk, inflammation, and infectious diseases has been reported by several researchers [6,7,8]. Vitamin D plays a key role as an immunomodulator, and its level has been associated with reduced severity of infection [9,10]. With emerging cases of the COVID-19 pandemic, it is now clear that there exists a close relationship between the SARS-CoV-2 virus and the immune system, resulting from the hyper-induction of pro-inflammatory cytokines [11]. In the literature, it has been established that vitamin D is able to modulate the immune responses against COVID-19 infection by alleviating the intensity of this cytokine storm [12,13,14]. In this context, there are several findings that associated lower levels of serum vitamin D with severity of infection in COVID-19 patients [15,16,17,18,19]. However, most of these published works only aimed to evaluate the serum level of 25-hydroxyvitamin D in patients. Our research strategy of simultaneous quantitation of vitamin D and its metabolites in dried blood spot (DBS) of patients who tested positive or negative for COVID-19 will contribute to the management of COVID-19 patients and establishment of potential treatment guidelines. Moreover, the level of vitamin D and its metabolites could be used as potential biomarker for assessing disease risk in patients with COVID-19 infection.

For the quantitation of vitamin D in research and clinical practice, liquid chromatography–tandem mass spectrometry (LC–MS/MS) has increasingly been used in terms of high specificity and sensitivity. Small endogenous molecules such as vitamin D can be assayed accurately by LC–MS/MS methods, and the technique only requires small sample volumes [20,21]. Another major advantage of LC–MS/MS analysis of vitamin D is that it can measure multiple low abundance vitamin D metabolites in the same analysis. Moreover, incorporating steps to derivatize vitamin D and its metabolites can greatly increase assay sensitivity [22,23]. In this study, DBS have been utilized for sample collection as an alternative to serum or plasma samples for the subsequent measurement of vitamin D and metabolites by LC–MS/MS. DBS is a micro-sampling method, requiring a small volume of blood sample (i.e., ≤50 µL), and these samples are easy to collect, store, and transport. The DBS were self-collected which can decrease the burden on clinical sites during pandemics. This technique also presents minimal invasiveness and is suitable for neonates and elderly patients, in whom blood volume or vascular access may be limited [24,25].

The principal aim of our study was to develop and validate a robust LC–MS/MS method for the simultaneous quantitation of two major forms of vitamin D (D2 and D3) and their 25-hydroxy metabolites in DBS. Our secondary objective was to assess the feasibility of this method to determine vitamin D concentrations easily and accurately in DBS as a potential biomarker for disease severity. To date, various LC–MS/MS methods have been described in the literature using 100–200 µL of biological matrix with a lower limit of detection of 5 ng/mL [26]. Here, our developed method could detect vitamin D and its metabolites in small volumes of human blood (~20 µL) concentrations from 0.78 ng/mL. We also summarized the comparison of the current method with the previously reported LC–MS/MS methods related to the quantitation of vitamin D in DBS and human plasma or serum in Appendix A. This validated method will facilitate the use of DBS for assessing blood concentrations of vitamin D and its metabolites, and can be easily implemented for the remote collection of specimens needed for the assessment of potential biomarkers.

## 2. Results and Discussion

### 2.1. Chromatographic and Mass Spectrometric Conditions Optimization

To optimize mass spectrometric conditions for the analysis of vitamin D3, vitamin D2, 25(OH)D3, 25(OH)D2 and stable deuterated isotope labeled internal standards (IS), including d-6 25(OH)D3 and d-3 vitamin D2, both electrospray ionization (ESI) and atmospheric pressure chemical ionization (APCI) conditions were assessed with and without derivatization. In positive electrospray ionization mode, all the analytes showed higher signal intensity and low signal-to-noise ratio (S/N) compared to APCI with 4-Phenyl-1,2,4-triazoline-3,5-dione (PTAD) derivatization. MS data for derivatized analytes are shown in Table in Section 3.3; MS data without derivatization are in the Appendix A. No endogenous interfering peaks were identified in the MS/MS chromatograms at the retention times of the analytes of interest after PTAD derivatization (Figure 1). To compare the value of obtained data collected in conventional MRM with scheduled multiple reaction monitoring (sMRM) mode, data were further analyzed, and it was found that sMRM mode was superior to the conventional MRM mode because MRM transition was monitored only during short retention time window, maximizing the dwell times.

Furthermore, without derivatization, the major sMRM fragments in mass spectra were as follows: 25(OH)D2: 413.15 > 395.4, 25(OH)D3: 401.15 > 383.4, vitamin D2: 397.3 > 69.1, vitamin D3: 385.3 > 367.4 (Appendix A). Compared to derivatized MRM fragments, derivatization shifted all precursor ions m/z by approximately 150 Da to the region of 560~600 Da (Table in Section 3.3), which reduces the chance of interference from low background interference. For the PTAD derivative of 25(OH)D2 and 25(OH)D3, there was elimination of one water molecule from the protonated molecule and (M+H-H_2_O)^+^ was used as the precursor ion. On the contrary, the PTAD derivative of vitamin D2 and D3 gave the precursor ion peak at 572.35 and 560.3, respectively (Appendix A). PTAD derivatization showed a major fragment at ~298 m/z ratio for all the analytes which confirmed better selectivity.

LC parameters were optimized using ACE Excel C18 PFP, 1.7 µm particle size, 2.1 × 100 mm column protected with a C18 guard column (Phenomenex, Torrance, CA, USA). Mobile phase conditions such as flow rate and composition were investigated to achieve higher ion intensity, shorter retention times, and sharper peak shape with no peak tailing. After experimenting with different solvents, 0.1% *v/v* formic acid in water was selected as mobile phase A and 0.1% *v/v* formic acid in methanol (MeOH) was selected as mobile phase B. The total flow was 0.5 mL/min. Formic acid is commonly used as a modifier for analysis of vitamin D, and has been shown to improve ionization intensity [27]. The column temperature was maintained at room temperature (RT) and injection volume was 10 µL. After evaluating different gradient profiles, all analytes were eluted within the 95–97% solvent B gradient window at 3 to 9 min. Here, more organic solvent B was needed for the elution of hydrophobic vitamin D analytes from the reversed-phase C18 column. Obtained retention times for 25(OH)D3, 25(OH)D2, vitamin D3, and vitamin D2 were 3.191, 3.243, 5.654, and 5.549 min, respectively. Selected IS, d-6 25(OH)D3 and d-3 vitamin D2 showed similar chromatographic behavior and elution time without prolonging the total run time. No interference from the endogenous components was observed in the plasma at analytes or IS retention time from the blank DBS chromatograms (Figure 1a).

### 2.2. Assay Validation

#### 2.2.1. Sensitivity and Selectivity

The selectivity of the method was determined by analyzing six blank blood samples from individual sources to determine if there was any interference at the retention time of the analyte and IS. At all the retention times of analytes, no significant interference was observed from any endogenous substance (Figure 1).

The sensitivity of the method was determined by the lower limit of quantitation, LLOQ (0.78 ng/mL) with 10:1 S/N. The assay was linear over the range of 0.78–200 ng/mL. The LLOQ had acceptable inter- and intraday accuracy and precision for all analytes (Table 1).

#### 2.2.2. Accuracy and Precision

The accuracy of the quantitative analysis of the analytes was within the acceptance limits of FDA guidelines at all concentration levels. The % bias of accuracy values ranged from −11.8 to 16.4%, within the acceptance limits at all concentration levels indicating acceptable assay accuracy. The accuracy and precision results are listed in Table 1. The % RSD of precision was within 4.5–14.4% indicating good precision values.

#### 2.2.3. Calibration Curve and Linearity

The calibration curves for vitamin D2 and vitamin D3 and their metabolites were linear covering the concentration range of 0.78 to 200 ng/mL for all analytes. The LLOQ of analytes was found to be 0.78 ng/mL, which was suitable for the plasma concentration determination of clinical samples.

#### 2.2.4. Carry-Over

After injecting high concentration QC samples, blank samples were analyzed, and no significant peaks (>20% of the LLOQ) were observed, which confirmed no significant carry-over effect was present.

#### 2.2.5. Recovery and Matrix Effect

Recovery was calculated by comparing the analyte at low quality control (LQC) and high quality control (HQC) response of spiked pre-extraction and post-extraction human DBS samples. The % extraction mean recovery for all analytes was approximately 50% and consistent across the QC concentration levels. The mean recovery of d6 25(OH)D3, d3-vitamin D2 was 56.6 ± 4.8 and 59.6 ± 3.1, respectively. The absolute mean recoveries are shown in Table 2. The matrix effect’s values were within ±15% range for all analytes, which indicated the matrix effect was not significant enough to interfere with the ionization.

### 2.3. Discussion

The LC–MS/MS technique has been considered as the gold standard for the quantitative analysis of vitamin D in plasma for more than a decade. LC–MS/MS offers improved specificity, sensitivity, and accuracy for vitamin D assay compared to the immunoassay method, and for that it is a more preferrable technique in research and clinical settings [22,28]. Nevertheless, the analytical method of vitamin D requires several factors to be considered: choice of biological matrix, sample preparation method, and extraction procedure. In these terms, our study demonstrates an efficient and robust LC–MS/MS method for the quantitation of vitamin D and its metabolites in plasma. We employed DBS as the sample matrix which required only 40 µL of the patient’s blood, ensuring the micro sampling technique. For the extraction of vitamin D and its metabolites, different extraction methods—solid phase extraction (SPE) and liquid liquid extraction (LLE)—were employed, followed by derivatization. Among SPE and LLE, SPE had better extraction recovery and thus it was chosen over LLE. Subsequently, HLB 96-well plates cartridges (30 mg; polymeric reversed-phase sorbent Waters Corporation, Milford, MA) were used in the SPE procedure and ensured enhanced retention of analytes with higher capacity and selectivity, and provided better clean up. SPE also has advantages over LLE in terms of low solvent consumption, time-saving, greater reproducibility, and automation [29]. The extraction methods of vitamin D and its metabolites from DBS samples were evaluated by assessing the absolute extraction recoveries (AER) with two concentrations (LQC and HQC) (Table 2). The DBS samples were first wet with 100 µL of water and then vortexed for 5 min followed by sonication. Agitation with water ensured initial extraction of vitamin D from the DBS. Ice-chilled methanol was further added to aggregate proteins by electrostatic interactions. In the literature, we have found that vitamin D and its metabolites remain bound to vitamin D binding protein (VDBP) and albumin in plasma [30]. Thus, for additional protein precipitation mechanisms, before adding MeOH, a salt zinc sulfate (0.2 mM) solution was mixed which would aggregate proteins due to the hydrophobic interactions, typically known as salting out [31]. Further samples were processed as per the SPE method described in 3.5.2, and eluents were dried under nitrogen.

After SPE, samples were derivatized by adding PTAD (50 µL, 0.1 g/L in anhydrous ethyl acetate) and vortexed for 30 min at RT. PTAD reacts with conjugated diene systems of vitamin D by Diels–Alder addition reaction [32]. Vitamin D and its metabolites are found at low levels in the biological samples and have low ionization efficiency. Thus, derivatization with PTAD was carried out to increase the ionization efficiency and sensitivity of the quantitation method. For reconstitution, we chose a 75:25 ratio of 0.1% formic acid in MeOH to 0.1% formic acid in water to be consistent with the mobile phase conditions.

Extraction recovery and matrix effect were consistent throughout the calibration range. The sample clean-up method was efficient and reproducible which diminishes the interference from other endogenous components. With the DBS sample, a total of approximately 10 µL blood was required for sample preparation. Nevertheless, extraction recovery was moderate, and, in addition, final quantitation was achieved with optimum selectivity and sensitivity. In addition, with PTAD derivatization better selectivity and low S/N were achieved, which confirmed the method’s suitability for the quantitation. Considering the LC parameters, our method was faster with a short analysis time, it was more sensitive for all vitamin D and analytes, and also showed a wide range of linearity.

### 2.4. Clinical Application of the Method for Pharmacokinetic Studies and Significance

The developed and validated LC–MS/MS method was successfully applied for the quantitation of vitamin D and its metabolites in 909 DBS samples collected from the participants tested for COVID-19. A total of 81 participants’ DBS samples were not sufficient to be analyzed. Among 828 clinical samples, using our validated method, we measured the blood concentrations of the major metabolite 25(OH)D3 at a range of 2–195.6 ng/mL, and none of them fell below the BQL (below quantitation level). Vitamin D3, 25(OH)D2, and vitamin D2 were quantitated at a range of 0.5–121.5 (208 samples < BQL), 0.6–54.9 (692 samples < BQL), and 0.5–23.9 ng/mL (736 samples < BQL), respectively. We will be briefly presenting the statistical significance and clinical relevance of these data to our future manuscript. In the literature, several studies reported the lowest limit of quantitation as ≤ 2.5 ng/mL for both 25(OH)D2 and 25(OH)D3, with a linearity range between 2.5 and 100 ng/mL [33,34]. Lote-Oke et al. (2020) determined the levels of vitamin 25(OH)D3 concentrations in human serum and DBS samples, and reported mean vitamin 25(OH)D3 concentrations as 47 ± 19 and 44 ± 19 ng/mL, respectively [26].

## 3. Materials and Methods

### 3.1. Chemicals and Materials

25(OH)D2, 25(OH)D3, vitamin D2, and vitamin D3, and IS, including d-6 25(OH)D3 and d-3 vitamin D2, were purchased from Cerilliant (Round Rock, TX, USA). Methanol (LC–MS grade MeOH), formic acid, and acetonitrile were purchased from Fisher Scientific (Fair Lawn, NJ, USA). Barnstead GenPure water purification systems from ThermoScientific, (Waltham, MA, USA) were used to generate ultrapure water. Whatman 903 protein saver cards and Centrifuge tubes were purchased from Cytiva- Little Chalfont, UK and Corning Co., Corning, NY, USA, respectively. The Oasis Prime HLB 96-well plate (30 mg/1 mL; Waters Corporation, Milford, MA, USA) was used for SPE. A BSD600 Duet device (Model No. BSD600/DUET, BSD Robotocs Brisbane, Australia) was used to collect a DBS punch for sample analysis. A Resprep VM-96 Vacuum Manifold for 96-well Plates (Restek Corporation, Bellefonte, PA, USA) was used for sample processing. A Biotage TurboVap 96 Concentration Evaporator Workstation, from Biotage (Uppsala, Sweden) was used for post-extraction sample drying. Human blood samples for preparation of DBS standards were obtained from the ZenBio (Durham, NC, USA).

### 3.2. Preparation of Vitamin D-Free Artificial Blood

Human blood was centrifuged at 664× *g* (2500 rpm) for 10 min at RT to separate the blood samples into plasma and cellular components. After centrifugation, collected plasma was aliquoted into separate tubes and preserved for preparation of vitamin D-free plasma. For separated cellular components, 25 mL 0.1 M phosphate-buffered saline (PBS) solution (pH 7.4) was used to wash isolated cellular components thoroughly, followed by centrifugation at 664× *g* for 10 min. This centrifugation step was repeated 5 times for the maximum removal of endogenous vitamin D. Subsequently, previously aliquoted plasma was mixed with activated charcoal (0.14 g charcoal/10 mL of plasma) by continuous shaking on a magnetic stirrer for 8 h. Following centrifugation at 12,298× *g* for 25 min, the supernatant was collected, and the centrifugation step was repeated, and the filtration step was repeated for the collection of remaining plasma as previously described [35]. This artificial vitamin D-free whole blood (10 mL) was subsequently used as a homogenous mixture of plasma and blood cells.

### 3.3. Liquid Chromatographic and Mass Spectrometric (LC–MS/MS) Conditions

A Shimadzu 8060 NX triple quadrupole mass spectrometer equipped with a dual ion source (DUIS), operated in positive ESI mode, was utilized for sample analysis. LabSolutions LC–MS software Version 5.99 SP2 from Shimadzu Scientific, Inc., (Columbia, MD, USA) was used to control the instrument and for data acquisition. MS parameters, such as temperature, voltage, gas pressure, etc., were optimized utilizing the software auto method optimization through a product ion scan with 1 µg/mL solution in MeOH. The chromatographic separation of vitamin D and its metabolites was performed using an ACE Excel C18 PFP column (1.7 µm particle size, 2.1 × 100 mm, MACMOD Analytical, Chadds Ford, PA, USA) protected with a C18 guard column (Phenomenex, Torrance, CA, USA). Mobile phase A (0.1% *v/v* formic acid in water; 25%) and mobile phase B (0.1% *v/v* formic Acid in MeOH; 75%) were operated in gradient mode with a flow rate of 0.5 mL/min and had the following profile: 0.1–1 min, 75% B; 1–3 min, 75–95% B; 3–6.49 min, 95% B; 7–9 min, 97% B followed by column re-equilibration with 75% solvent B from 9.1 to 11 min.

The sMRM transitions for each vitamin D species and IS are represented in Table 3, including their respective optimum MS parameters such as voltage potential (Q1, Q3) and collision energy (CE).

### 3.4. Preparation of Stock, Calibration Standards, and Quality Control Samples

Original stock solutions of vitamin D3, vitamin D2, 25(OH)D2, and 25(OH)D3 were mixed to prepare working stock solutions in MeOH and were stored frozen at −20 °C in aliquots. The calibration curves covered the concentration ranges from 0.78 to 200 ng/mL for all analytes.

Two stable isotope-labeled IS (d-6 25(OH)D3 and d-3 vitamin D2) were mixed for the final concentration of 200 ng/mL. Concentrations selected for quality controls were as follows: lower limit of quantification (LLOQ; 0.78 ng/mL), low quality control (LQC; 3.12 ng/mL), middle quality control (MQC; 25 ng/mL), and high-quality control (HQC; 150 ng/mL).

### 3.5. Preparation of Artificial Blood Spiked and DBS Samples

#### 3.5.1. Preparation of Blood and DBS Cards

To prepare the DBS card, the vitamin D-free artificial blood, as described above (98 µL), was first spiked with appropriate calibration and QCs stock (2 µL) and the spiked blood was left for 3 min at 37 °C temperature to reach red blood cells (RBC)/plasma equilibrium and to mimic RBC/plasma analyte distribution. DBS cards were prepared by pipetting 40 µL of the spiked blood (sample) or the vitamin D-free artificial blood (blank) on each spot on the Whatman 903 protein saver cards. The prepared DBS card was then kept in the dark at RT for 3 h to facilitate drying. Subsequently, the card was kept in light protective Whatman plastic ziplock bags (GE Healthcare Life Sciences) using a suitable desiccant, and stored at –2 °C until further punching and extraction.

#### 3.5.2. DBS Sample Extraction

For each DBS sample, four 3 mm diameter punches were obtained using a BSD Duet device. The DBS punched sample was collected in a 96-well plate and the sample was hydrated with 100 µL water and 10 µL of mix IS solution (d-6 25-(OH) D3 and d-3 vitamin D2; 0.2 µg/mL). After addition, the plates were vortexed for 5 min on a Mixmate (Eppendorf, North America) at 1200 rpm and then sonicated for 10 min. After that, a solution of ZnSO_4_ (400 µL of 0.2 mM) was added in each well, vortexed for 5 min on a Mixmate at 1200 rpm, then sonicated for 30 min at RT. Ice chilled MeOH (400 µL) was added to each well and sonicated for 15 min at RT, and then the sample underwent solid phase extraction (SPE). First, MeOH (200 µL) was added to each well on the 96-well plate (HLB, Waters, Millford, MA), and eluted to waste. Next, water (200 µL) was added to each well for conditioning, was eluted, and then previously prepared samples were loaded to the cartridges and samples were subjected to washing using MeOH (200 µL 60%) two times. For final elution, 300 µL MeOH was passed through the cartridges two times and eluent was collected into 96-well plates. Collected eluent was then dried using a nitrogen evaporator and sample derivatization was performed. Briefly, the above dried sample PTAD (4-Phenyl-1,2,4-triazoline-3,5-dione) (50 µL of 0.1 g/L) in anhydrous ethyl acetate was added and vortexed for 30 min at 500 rpm on Mixmate. The sample was evaporated under nitrogen, and MeOH (100 µL with 0.1% formic acid) was added to the dried residue for reconstitution. The reconstituted solution was vortexed on a Mixmate at 1200 rpm for 1 min, followed by sonication for 2 min in a sonicator. Afterwards, formic acid (25 µL 0.1% in water) was added, and the sample was centrifuged at 3000 rpm for 5 min. A 100 µL aliquot of the clear supernatant was transferred to a 96-well autosampler plate, and 10 µL were injected into the LC–MS/MS system for analysis.

### 3.6. Method Validation

The method was validated in DBS samples following the standards mentioned in the 2018 Food and Drug Administration (FDA) guidelines for Bioanalytical Method Validation [36].

The sensitivity of the method was determined from the S/N of the response of the analyte in calibration standards. By calculating the S/N of the analyte response in the calibration standards, method sensitivity was assessed. The threshold value for S/N was required to be greater than three for the lower limit of detection (LOD) and greater than ten for the LLOQ for all analytes.

Calibration curves were generated utilizing concentration and response data and weighted least-squares linear regression analysis (1/x^2^). Integration of peak areas, the analyte/IS peak area ratio, and the concentration determination of each standard were conducted using the LabSolutions LC–MS software Version 5.99 SP2. The accuracy acceptance criteria were required to be within ±15% of the nominal concentration for all non-zero calibrators, and within ±20% for the LLOQ level in each validation run. To assess accuracy and precision, QC samples were prepared at four concentration levels, LLOQ, LQC, MQC, and HQC and analyzed. To determine both intra- and inter-day accuracy and precision, each QC sample was prepared in five replicates of different blood sources and analyzed in three separate validation runs over three days. Acceptance criteria for both accuracy and precision were required to be within ±15% for each concentration level except for the LLOQ, where ±20% was deemed acceptable.

To estimate the carry-over effect, a blank sample was subsequently injected into the LC–MS system after running an HQC, and the response from the blank sample was measured. Carry-over was required to be below 20% of the response of a processed LLOQ sample.

Extraction recoveries of analytes and labeled IS from charcoal-stripped surrogate and normal blood were determined by dividing the peak area ratio of analyte to IS (after subtracting any endogenous background) from post-spiked samples and multiplying by 100% for both the low and high QCs (n = 3). The matrix effect was measured by dividing the peak area of the post-spiked samples by the peak area of the sample in neat solvent.

### 3.7. Study Design and Blood Collection

DBS samples were collected from the participants who volunteered for PK sub-studies within two separate clinical trials investigating HCQ (hydroxychloroquine) as a post-exposure prophylaxis (PEP) (ClinicalTrials.gov; NCT04328961) [37] or treatment of COVID-19 (ClinicalTrials.gov; NCT04354428) [38]. Participants who contributed at least three DBS were included in the analysis. The PEP study was conducted in participants identified as close contacts of a person with a PCR-identified positive test for SARS-CoV-2. Participants signed written informed consent, and procedures were approved by the Western Institutional Review Board. All the procedures involving handling human DBS samples were approved by the University of Nebraska Medical Center (UNMC) Institutional Biosafety Committee (IBC protocol number 20-04-025-BL1). Whole blood for DBS sampling was obtained through venipuncture or capillary blood using a sterile finger pricker to puncture the tip of the finger. Approximately 100 μL of blood was applied to the DBS card and then left to dry for at least 3 h. All the samples were stored at −20 °C at the clinical trial site. All samples were shipped to UNMC, Omaha, NE, USA, with dried ice and stored at −80 °C until processed for analysis. The vitamin D2 and D3 and their metabolites were quantified in DBS samples with this simultaneous validated LC–MS/MS method. The statistical analysis of the validation results was performed using Microsoft Office Excel^®^ 365.

## 4. Conclusions

We developed and validated a robust, sensitive, and reproducible LC–MS/MS method for quantitating 25(OH)D3, vitamin D3, 25(OH)D2, and vitamin D2 in human blood, utilizing a DBS sample collection technique. Our developed method was advantageous with simple sample collection, limited amount of blood sample required, easy transfer and storage conditions, and it exhibited excellent drug stability. The LC–MS/MS bioanalytical method was validated according to FDA guidelines. The method showed superior selectivity and sensitivity, with nominal concentrations as low as 0.78 ng/mL. The method showed no significant matrix effects, and acceptable accuracy and precision that fulfilled the FDA requirements.

## Figures and Tables

**Figure 1 ijms-24-05489-f001:**
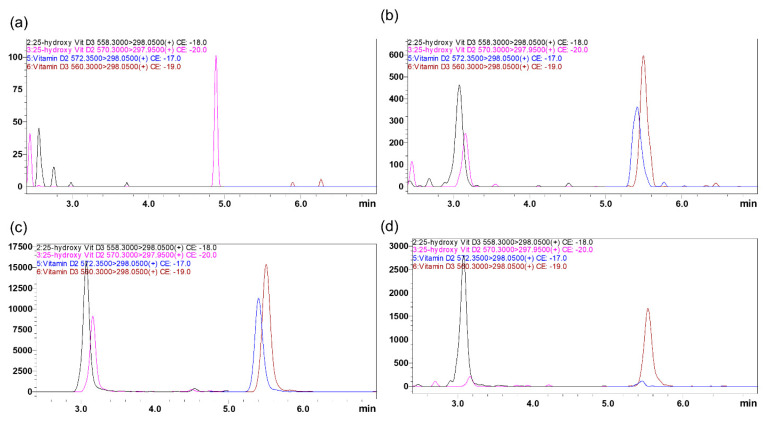
Representative MRM ion-overlay chromatograms of vitamin D2 and D3 and their 25-hydroxy metabolites extracted from DBS prepared from vitamin D free artificial blood (**a**) blank, and with standard spiked at LQC (**b**) and MQC level (**c**). (**d**) Clinical Sample(#4200007901B2) extracted from DBS.

**Table 1 ijms-24-05489-t001:** Assessment of accuracy (% Bias) and precision (% R.S.D.) of 25(OH)D2, 25(OH)D3, vitamin D2, and vitamin D3 in human DBS.

Analyte	QCs	Accuracy (% Bias)		Precision (% RSD)	
	Levels	Intra Day	Inter Day	Intra Day	Inter Day
25(OH)D2	LLOQ	3.3	5.1	11.8	13.0
	LQC	2.4	1.7	7.7	7.0
	MQC	8.7	4.4	10.9	13.2
	HQC	7.2	10.2	12.6	6.0
25(OH)D3	LLOQ	11.4	14.1	1.9	4.5
	LQC	5.1	4.8	4.7	6.7
	MQC	−2.5	−2.2	2.8	11.3
	HQC	11.1	6.3	9.1	7.4
vitaminD2	LLOQ	10.7	16.4	6.2	14.4
	LQC	−1.9	−5.0	6.4	10.3
	MQC	8.7	2.3	6.8	10.7
	HQC	−3.0	−10.0	0.9	7.0
vitaminD3	LLOQ	15.0	9.5	7.7	4.7
	LQC	3.3	−3.2	7.9	7.4
	MQC	1.0	1.7	5.4	13.5
	HQC	−3.4	−11.8	0.4	12.1

**Table 2 ijms-24-05489-t002:** The extraction recovery (%) of 25(OH)D2, 25(OH)D3, vitamin D2, and vitamin D3 after pre- and post-extraction spike of human DBS. Values are given as a mean value ± SD of three experiments.

Analyte	Extraction Recoveries (Mean ± SD, n#3) with (%)	
	LQC	HQC
25(OH)D2	57.2 ± 5.3	51.4 ± 8.3
25(OH)D3	53.7 ± 7.4	55.05 ± 7.7
vitamin D2	48.4 ± 5.05	53.4 ± 7.5
vitamin D2	57.4 ± 3.99	60.1 ± 6.8

**Table 3 ijms-24-05489-t003:** Summary of MS/MS parameters with 4-Phenyl-1,2,4-triazoline-3,5-dione (PTAD) derivatization: precursor ion, fragment ions, voltage potential (Q1), collision energy (CE), and voltage potential (Q3), retention time, and linearity for analytes.

Analyte	MRM (Precursor/Fragment Ions)	Q1 (V)	CE (V)	Q3 (V)	Retention Time (min)	Linearity (ng/mL)
25(OH)D2	570.30 > 297.95	−40.0	−29.0	−22.0	3.243	0.78–200
25(OH)D3	558.30 > 298.05	−11	−21	−18	3.191	0.78–200
Vitamin D2	572.35 > 298.05	−10	−21	−12	5.549	0.78–200
Vitamin D3	560.30 > 298.05	−11	−22	−13	5.654	0.78–200
d-6 25(OH)D3	564.30 > 298.00	−19	−25	−13	3.168	NA
d-3 vitamin D2	575.35 > 301.10	−12	−22	−18	5.5	NA

## Data Availability

The authors state that all the data related to the findings of this study are available in the article and the Appendix A. Additional data can be provided on request.

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
