# Peer review of "A Simultaneous Extraction/Derivatization Strategy for Quantitation of Vitamin D in Dried Blood Spots Using LC–MS/MS: Application to Biomarker Study in Subjects Tested for SARS-CoV-2"

_ijms, 2023, doi:10.3390/ijms24065489_

Round 1
Reviewer 1 Report
The manuscript by Chhonker et al. examined a simultaneous extraction/derivatization strategy for quantitation of Vitamin D in dried blood spots using LC-MS/MS. They found that our developed LC−MS/MS method may be used for quantification of Vitamin D and its metabolites in DBS and may be applied to investigations of the emerging role of these compounds in various physiological processes.
The major conclusions of this research are justified by the results. The methodology seems to be correct in most experiments and the results of this work may be worth publishing. However, the study requires improvement in some aspects. Please consider the following points:
Minor revision:
1. In figure 1, the figures were not very clear, please replace them.
2. In line 330, the authors described that the absolute mean recoveries are shown in Table 6, but where is table 6?
3. Did the authors compare this method with the published methods?
4. Did the authors use the clinical samples to compare the accuracy of this method and other published methods? How many clinical samples were used?
Author Response
The manuscript by Chhonker et al. examined a simultaneous extraction/derivatization strategy for quantitation of Vitamin D in dried blood spots using LC-MS/MS. They found that our developed LC−MS/MS method may be used for quantification of Vitamin D and its metabolites in DBS and may be applied to investigations of the emerging role of these compounds in various physiological processes.
The major conclusions of this research are justified by the results. The methodology seems to be correct in most experiments and the results of this work may be worth publishing. However, the study requires improvement in some aspects. Please consider the following points:
Minor revision:
- In figure 1, the figures were not very clear, please replace them.
Reply: It has been replaced by the high-quality image
- In line 330, the authors described that the absolute mean recoveries are shown in Table 6, but where is table 6?
Reply: It is typological error and has been corrected to table 3.
- Did the authors compare this method with the published methods?
Reply: Provided in the introduction and tabular form of comparison of previously reported LC-MS/MS methods related to the quantitation of vitamin D in dried blood spots or human serum included in supplementary material table S2.
- Did the authors use the clinical samples to compare the accuracy of this method and other published methods? How many clinical samples were used?
Reply: 909 clinical samples were used in our analysis. In section 3.8 we have included the conc. range of all analytes found in the clinical samples, which were within the calibration range and also compared with other published methods. We will briefly present the statistical significance and clinical relevance of this data to our future manuscript. In this one, we have highlighted the methodology part.
Reviewer 2 Report
Reviewer comments:
In the manuscript (IJMS-2210678) the authors have developed and validated according to FDA guidelines an efficient and robust LC-MS/MS method for the simultaneous quantitation of vitamin D2 and vitamin D3 and their 25-hydroxy metabolites in human blood. The advantage of applied LC-MS/MS method for the determination of vitamin D is its better specificity, sensitivity and accuracy for vitamin D assay compared to immunoassay method. Beside this, this method used DBS as the sample matrix which used only 40 ul of patient’s blood ensuring the micro sampling technique.
The manuscript can be very interesting for readers, because it offers the new method of vitamin D quantification which is very sensitive with LLOD of 0.78 ng/mL. This is very important because in real biological samples the concentrations of individual vitamin D metabolites are common below the quantitation level obtained for other developed methods.
In this form, the manuscript needs several minor corrections before the final decision. Please, take into account below some of the comments and suggestions for the improvement of your manuscript quality. In the pdf file of the manuscript the text for the correction has been marked.
Title page
Lines 6-11 and 16-21
Please, replace bold text with normal. Correct font size in E-mail.
Line 29 and in the other part of manuscript
According to the WHO, abbreviation for the coronavirus disease 2019 is COVID-19.
Line 32
Replace Formic Acid with formic acid
Line 33
Replace ml with mL. Delete s in techniques.
Line 37
In a sentence, ng/mL can be omitted after the first three numbers.
Lines 39, 99
Replace V with v in Vitamin D.
Lines 46-49 and in the rest of the manuscript.
Individual forms of vitamin D like D2, D3 should be written as D2 and D3.
Line 49
Please, add s to the form.
Line 51-57
Please replace the text with:
The first biotransformation reaction occurs in the liver where vitamins D2 and D3 are converted by the hydroxylation reaction into corresponding 25-hydroxy metabolites (25(OH)D3 or 25(OH)D2)). The reaction is catalysed by two isoforms of microsomal cytochrome p450 (CYP27A1 and CYP2R1). In the next step, in the renal proximal tubule of the kidney, corresponding 25-hydroxy metabolites of vitamins D2 and D3 are converted into 1,25-dihydroxy or 24,25-dihydroxy metabolites by the action of microsomal enzymes CYP27B1 or CYP24A1, respectively .
Line 57
Please, insert possible dihydroxy between two metabolites. Add of vitamin D after metabolites.
Line 77
Please, replace participants tested for Covid-19 and negative Covid-19 with patients positive or negative on COVID-19
Line 89
Add its after and
Line 91
Replace l with L.
Line 94
Delete full stop before references.
Line 96
Please, replace the quantitation of vitamin D and primary metabolites with the simultaneous quantitation of two major forms of vitamin D (D2 and D3) and their 25-hydroxy metabolites in DBS.
Material and methods
Line 104 and in the rest of manuscript
Please replace Vit D2 or Vit D3 with vitamin D2 or vitamin D3.
Line 106
Please, insert (MeOH) after Methanol.
Line 119
Insert space after 664.
Lines 120-122 and in other part of text
Please, replace serum with plasma. For this experiment human blood with anticoagulant was probably used, so after that you can only get plasma and different blood cells.
Line 122
Please, insert concentration of buffer.
Line 126
Delete m in gm
Line 142
Insert its after and.
Line 143 and line 277
Insert m after µ, replace 2.1* 100mm, with 2.1x 100 mm,
Lines 145 and 146
MeOH and formic acid instead of methanol and Formic Acid.
Line 147
Insert B after %.
Table 1
Please, correct the table.
Each analyte in one row.
Vit D2 and D3 replace with vitamin D2 and Vitamin D3. Use subscript for the number of vitamins. Retention Time replace with Retention time (min), MRM replace with MRM (precursor/fragment ions)
Insert * 4-Phenyl-1,2,4-triazoline-3,5-dione (PTAD) below the table
Lines 158-160
Please delete this sentence, the same sentence is in line 104-106.
Line 160-162
Please, replace ….. each analyte with Orginal stock solutions of vitamin D3, vitamin D2, 25(OH)D2 and 25(OH)D3,were mixed….
Line 162
Insert curves after calibration
Line 169
Please replace title with
Preparation of artificial blood spiked and DBS samples
Line 170
New title Preparation of blood and DBS cards
Line 173
Replace RBC with red blood cells (RBC). Delete - in vivo.
Line 175
Insert (sample) or the vitamin D free artificial blood (blank) between blood and on.
Line 181, 186
Insert space
Line 185
Insert After that, solution of before ZnSO4
Line 198
Replace Fa with formic acid
Line 201
Delete ) after % and insert after water.
Line 235
Formatting
Line 248
Replace hr with hours.
Line 250
Replace D with D2 and D3
Line 251
Please, insert statistical analysis.
Results and discussion
Line 255
Delete internal standards and parentheses.
Line 258
Replace signal-to noise ratio with S/N.
Line 263
Delete (scheduled multiple reaction monitoring).
Line 262
Did you think the next:
To compare the value of obtained data collected in conventional MRM with sMRM mode, data were further analysed and it found that sMRM mode was superior to the conventional MRM mode, because MRM transition was monitored only during short retention time window maximizing the dwell times.
Lines 269 and 270
Inser (TableS1) after 367.4. Insert (Table 1) after Da.
Line 285
Please, use room temperature or RT in the manuscript . Please look at lines 187 188 and other parts of the manuscript where the room temperature is mentioned.
Line 288
Please, explain why there are differences between obtained RT of analytes in the text (Line 289), Table 1 and Figure 1. Image 1 should be in a higher resolution. Supplementary data add chromatograms of IS, as well as at least several clinical samples which contain different levels of vitamins D and their metabolites.
Line 299
Insert a after 1.
Line 301
Replace signal-to-noise ratio with S/N.
Lines 304-305
Suggestion of title for Figure 1.
Representative MRM ion-overlay chromatograms of vitamin D2 and D3 and their 25-hydroxy metabolites extracted from DBS prepared from vitamin D free artificial blood (a) blank, and with standard spiked at LQC (b) and MQC level (c). (d) Clinical Sample(#4200007901B2) extracted from DBS.
Table 2
Correct name of analyte. Put levels in the second row, add y after da.
Line317
Replace D and with D2 and vitamin D3 and their
Line 319
Space and mL.
Line 330
Replace Table 6 with Table 3.
Line 333
Insert in the title of table recovery obtained for pre- or post extraction spike, Which of them?
Suggested title for the table 3
The extraction recovery (%) of 25(OH)D2, 25(OH)D3, vitamin D2 and vitamin D3 after pre- or post extraction spike of human DBS. Values are given as a mean value±SD of three experiments.
Table 3
In table 3 replace (Mean ± SD, n#3) with (%), delete % in front of Extraction. Check which data are related to vitamin D2 and vitamin D3.
Line 357
Delete space in ice-chilled
Lines 357-395
Please correct marked text according to the previous comments.
Line 384
Insert space before DBS.
Line 394
Insert (2020) after et al.
References
Please, check the font style.
Supplementary data
Please correct Supplementary Table 1 and Titles of figure 1 and figure 2.

Author Response
In the manuscript (IJMS-2210678) the authors have developed and validated according to FDA guidelines an efficient and robust LC-MS/MS method for the simultaneous quantitation of vitamin D2 and vitamin D3 and their 25-hydroxy metabolites in human blood. The advantage of applied LC-MS/MS method for the determination of vitamin D is its better specificity, sensitivity and accuracy for vitamin D assay compared to immunoassay method. Beside this, this method used DBS as the sample matrix which used only 40 ul of patient’s blood ensuring the micro sampling technique.
The manuscript can be very interesting for readers, because it offers the new method of vitamin D quantification which is very sensitive with LLOD of 0.78 ng/mL. This is very important because in real biological samples the concentrations of individual vitamin D metabolites are common below the quantitation level obtained for other developed methods.
In this form, the manuscript needs several minor corrections before the final decision. Please, take into account below some of the comments and suggestions for the improvement of your manuscript quality. In the pdf file of the manuscript the text for the correction has been marked.
Title page
Lines 6-11 and 16-21
Please, replace bold text with normal. Correct font size in E-mail.
Reply: Corrected
Line 29 and in the other part of manuscript
According to the WHO, abbreviation for the coronavirus disease 2019 is COVID-19.
Reply: Corrected
Line 32
Replace Formic Acid with formic acid
Reply: Corrected
Line 33
Replace ml with mL. Delete s in techniques.
Reply: Corrected
Line 37
In a sentence, ng/mL can be omitted after the first three numbers.
Reply: Corrected
Lines 39, 99
Replace V with v in Vitamin D.
Reply: Corrected
Lines 46-49 and in the rest of the manuscript.
Individual forms of vitamin D like D2, D3 should be written as D2 and D3.
Reply: In the literature vitamin D2 and D3 had been reported without a subscript. We would like to follow this pattern.
Line 49
Please, add s to the form.
Reply: Corrected
Line 51-57
Please replace the text with:
The first biotransformation reaction occurs in the liver where vitamins D2 and D3 are converted by the hydroxylation reaction into corresponding 25-hydroxy metabolites (25(OH)D3 or 25(OH)D2)). The reaction is catalysed by two isoforms of microsomal cytochrome p450 (CYP27A1 and CYP2R1). In the next step, in the renal proximal tubule of the kidney, corresponding 25-hydroxy metabolites of vitamins D2 and D3 are converted into 1,25-dihydroxy or 24,25-dihydroxy metabolites by the action of microsomal enzymes CYP27B1 or CYP24A1, respectively.
Reply: Thank you for your suggestion. Correction has been done with track changed.
Line 57
Please, insert possible dihydroxy between two metabolites. Add of vitamin D after metabolites.
Reply: Corrected
Line 77
Please, replace participants tested for Covid-19 and negative Covid-19 with patients positive or negative on COVID-19
Reply: Corrected
Line 89
Add its after and
Reply: Corrected
Line 91
Replace l with L.
Reply: Corrected
Line 94
Delete full stop before references.
Reply: Corrected
Line 96
Please, replace the quantitation of vitamin D and primary metabolites with the simultaneous quantitation of two major forms of vitamin D (D2 and D3) and their 25-hydroxy metabolites in DBS.
Reply: Corrected
Material and methods
Line 104 and in the rest of manuscript
Please replace Vit D2 or Vit D3 with vitamin D2 or vitamin D3.
Reply: Corrected
Line 106
Please, insert (MeOH) after Methanol.
Reply: Corrected
Line 119
Insert space after 664.
Reply: Corrected
Lines 120-122 and in other part of text
Please, replace serum with plasma. For this experiment human blood with anticoagulant was probably used, so after that you can only get plasma and different blood cells.
Reply: Corrected. Thank you for the clarification.
Line 122
Please, insert concentration of buffer.
Reply: Inserted
Line 126
Delete m in gm
Reply: Corrected
Line 142
Insert its after and.
Reply: Corrected
Line 143 and line 277
Insert m after µ, replace 2.1* 100mm, with 2.1x 100 mm,
Reply: Corrected
Lines 145 and 146
MeOH and formic acid instead of methanol and Formic Acid.
Reply: Corrected
Line 147
Insert B after %.
Reply: Corrected
Table 1
Please, correct the table.
Each analyte in one row.
Vit D2 and D3 replace with vitamin D2 and Vitamin D3. Use subscript for the number of vitamins. Retention Time replace with Retention time (min), MRM replace with MRM (precursor/fragment ions)
Insert * 4-Phenyl-1,2,4-triazoline-3,5-dione (PTAD) below the table
Reply: Corrected.
Lines 158-160
Please delete this sentence, the same sentence is in line 104-106.
Reply: Deleted
Line 160-162
Please, replace ….. each analyte with Original stock solutions of vitamin D3, vitamin D2, 25(OH)D2 and 25(OH)D3,were mixed….
Reply: Corrected
Line 162
Insert curves after calibration
Reply: Corrected
Line 169
Please replace title with
Preparation of artificial blood spiked and DBS samples
Reply: Corrected
Line 170
New title Preparation of blood and DBS cards
Reply: Corrected
Line 173
Replace RBC with red blood cells (RBC). Delete - in vivo.
Reply: Corrected
Line 175
Insert (sample) or the vitamin D free artificial blood (blank) between blood and on.
Reply: Corrected
Line 181, 186
Insert space
Reply: Corrected
Line 185
Insert After that, solution of before ZnSO4
Reply: Corrected
Line 198
Replace Fa with formic acid
Reply: Corrected
Line 201
Delete ) after % and insert after water.
Reply: Corrected
Line 235
Formatting
Reply: Corrected the formatting.
Line 248
Replace hr with hours.
Reply: Corrected
Line 250
Replace D with D2 and D3
Reply: Corrected
Line 251
Please, insert statistical analysis.
Reply: Included in the text.
Results and discussion
Line 255
Delete internal standards and parentheses.
Reply: Corrected
Line 258
Replace signal-to noise ratio with S/N.
Reply: Corrected
Line 263
Delete (scheduled multiple reaction monitoring).
Reply: Corrected
Line 262
Did you think the next:
To compare the value of obtained data collected in conventional MRM with sMRM mode, data were further analysed and it found that sMRM mode was superior to the conventional MRM mode, because MRM transition was monitored only during short retention time window maximizing the dwell times.
Reply: Thank you for the writing suggestion.
Lines 269 and 270
Inser (TableS1) after 367.4. Insert (Table 1) after Da.
Reply: Corrected
Line 285
Please, use room temperature or RT in the manuscript. Please look at lines 187 188 and other parts of the manuscript where the room temperature is mentioned.
Reply: Corrected all parts of the manuscript.
Line 288
Please, explain why there are differences between obtained RT of analytes in the text (Line 289), Table 1 and Figure 1. Image 1 should be in a higher resolution. Supplementary data add chromatograms of IS, as well as at least several clinical samples which contain different levels of vitamins D and their metabolites.
Reply: Thank you for the suggestion. RT has been corrected in the text. In figure 1, chromatogram of one clinical sample has already been included.
Line 299
Insert a after 1.
Reply: Corrected
Line 301
Replace signal-to-noise ratio with S/N.
Reply: Corrected
Lines 304-305
Suggestion of title for Figure 1.
Representative MRM ion-overlay chromatograms of vitamin D2 and D3 and their 25-hydroxy metabolites extracted from DBS prepared from vitamin D free artificial blood (a) blank, and with standard spiked at LQC (b) and MQC level (c). (d) Clinical Sample(#4200007901B2) extracted from DBS.
Reply: Corrected
Table 2
Correct name of analyte. Put levels in the second row, add y after da.
Reply: Corrected
Line317
Replace D and with D2 and vitamin D3 and their
Reply: Corrected
Line 319
Space and mL.
Reply: Corrected
Line 330
Replace Table 6 with Table 3.
Reply: Corrected
Line 333
Insert in the title of table recovery obtained for pre- or post extraction spike, Which of them?
Reply: In the methods part we have the extraction recovery calculation method. It is the peak area ratio of the pre-extraction spiked analyte to post-extraction spiked analyte.
Suggested title for the table 3
The extraction recovery (%) of 25(OH)D2, 25(OH)D3, vitamin D2 and vitamin D3 after pre- or post extraction spike of human DBS. Values are given as a mean value±SD of three experiments.
Reply: Corrected
Table 3
In table 3 replace (Mean ± SD, n#3) with (%), delete % in front of Extraction. Check which data are related to vitamin D2 and vitamin D3.
Reply: Corrected
Line 357
Delete space in ice-chilled
Reply: Corrected
Lines 357-395
Please correct marked text according to the previous comments.
Reply: Corrected
Line 384
Insert space before DBS.
Reply: Corrected
Line 394
Insert (2020) after et al.
Reply: Corrected
References
Please, check the font style.
Supplementary data
Please correct Supplementary Table 1 and Titles of figure 1 and figure 2.
Round 2
Reviewer 1 Report
no